# Surface Properties and Denitrification Performance of Impurity-Removed Rare Earth Concentrate

**DOI:** 10.3390/ma13030580

**Published:** 2020-01-26

**Authors:** Kai Zhang, Yuze Bai, Zhijun Gong, Zengwu Zhao, Baowei Li, Wenfei Wu

**Affiliations:** 1College of Environment and Energy, Inner Mongolia University of Science and Technology, Baotou 014000, Inner Mongolia, China; btzk@imust.cn (K.Z.); byz1107@163.com (Y.B.); Gong@imust.cn (Z.G.); lbw@imust.cn (B.L.); 2Key Lab Laboratory of Integrated Exploitation of Bayan Obo Multi-Metal Resources, Baotou 014000, Inner Mongolia, China; zze@imust.cn

**Keywords:** rare earth concentrate, active powder, structure–activity relationship, NH_3_-SCR, catalytic activity

## Abstract

Acid leaching and alkali roasting were used to remove impurities such as Ca and Si in Baiyun Obo rare earth concentrate. The effects of acid–base treatment on the physical and chemical properties of the samples were analyzed by scanning electron microscopy, X-ray diffraction, Brunauer–Emmett–Teller characterization, X-ray photoelectron spectroscopy, H_2_-temperature-programmed reduction, NH_3_-temperature-programmed desorption (TPD), and NO-TPD. Results showed that the content of Ce_7_O_12_ in the rare earth concentrates increased and the dispersion was uniform. The grains became smaller, the specific surface area of rare earth concentrates increased, and the active sites were more exposed. Ce coexisted in the form of Ce^3+^ and Ce^4+^, whereas Fe coexisted in the form of Fe^3+^ and Fe^2+^. The content of Fe^3+^ was increased. The acid–base-treated rare earth concentrates had a denitration efficiency of 87.4% at a reaction temperature of 400 °C.

## 1. Introduction

Rare earth-doped oxides are the main active constituents of catalysts for selective catalytic reduction with NH_3_ (NH_3_-SCR). The low-temperature activity and antitoxicity of rare earth element-doped transition metal elements are hot topics in the research field of novel SCR catalysts [1,2,3]. The rare earth concentrate contains large amounts of elements such as Ce, Fe, Mn, etc. Rare earth element and transition metal element-doped oxide can improve low-temperature SCR catalytic activity and its sulfur resistance. The catalytic activity and sulfur resistance of the catalyst are improved because of the abundant active sites on the surface and the coexistence of elements such as Fe and Ce in different valence states. The catalyst can increase the active sites and oxygen storage capacity and promote the reaction of gases such as NH_3_ and NO_X_ [4,5,6].

Natural minerals are used to improve the catalytic performance of minerals via physical and chemical methods; they are utilized as catalysts instead of those prepared by doping different elements synthesized using purified chemicals [7]. For example, the V_2_O_5_-WO_3_ (MoO_3_)/TiO_2_ catalyst prepared by using pure substances has a poor treatment effect on exhausted flue gas under low-temperature conditions [8,9,10]. According to the results of previous studies, the short plates of purely prepared catalysts are gradually being exposed. Therefore, developing new green and highly efficient catalysts by loading transition metal elements with natural minerals has become a topic of interest.

The Baiyun Obo fluorocarbon antimony ore has many complex mineral phases. The elements are fine and heterogeneous, and their natural mineral phases are complex and variable. In preparing catalysts using pure substances, various mineral phase structures coexist, and a plurality of elements cooperate to improve the catalytic activity of the catalyst [11,12]. In this study, we retained the natural mineral phase and modified and optimized the natural mineral phase structure to enhance the catalytic activity of natural minerals. In preparing the catalyst, the purified substances in the natural minerals do not need to be extracted. This avoids the complicated extraction process, and the catalyst ore phase prepared by using natural minerals is relatively stable compared with the catalyst synthesized using pure substances. A suitable liquid phase system was designed to dissolve it. The mineral phase, which favors the catalytic reaction, was exposed on the surface of the mineral, and a pore structure formed by dissolution and calcination was established on the surface of the mineral [13,14,15,16]. By re-integrating rare earth and transition elements, a solid solution, which is favorable for catalysis, was formed and was more exposed to the surface of the pores, thereby enhancing the dispersibility and acidic sites of the surface active components. The adsorption characteristics lowered the activation energy of the denitration reaction and the reaction temperature window of the catalyst prepared by the rare earth concentrate, thereby improving the catalytic activity of the catalyst [17,18,19]. These findings will have important theoretical significance and practical value in the research and development of new low-temperature SCR catalysts and play a promising role in the application of Baiyun Obo rare earth concentrates to industrial production.

## 2. Experimental

### 2.1. Materials

The main raw materials used in this experiment were 300–400-mesh rare earth concentrate powders produced from the No. 2 mine mouth of Baotou Baiyun Obo Mining Area (Baotou, China). Other reagents were of analytical grade.

### 2.2. Analysis and Testing Equipment

The test equipment included an X-ray diffractometer (PANalytical B.V., Amsterdam, Holland), a Sigma-500 field emission scanning electron microscope (Zeiss, Oberkochen, Germany), a STA449C thermal analyzer (Netzsch, Selb, Germany), a 3H-2000PS1 automated specific surface and aperture analyzer (BeiShiDe Instruments, Beijing, China), a PCA-1200 temperature-programmed chemical adsorption instrument (Builder, Beijing, China), a VERTEX70 Fourier in situ infrared spectrometer (Bruker, Billerica, MA, USA), an X-ray fluorescence analysis device (U-2200 RoHS Heavy Metal Detection Spectrometer), and a Thermo ESCALAB 250Xi X-ray photoelectron spectrometer (Thermo Fisher Scientific, Shanghai, China).

### 2.3. Sample Preparation

At a 300–400-mesh particle size, a certain amount of rare earth concentrates was crushed, ground, sieved, and dried.

a. Sample 1:

The rare earth concentrate raw ore was placed in an oven at 80 °C for drying, and after the water evaporated, the mineral on the filter paper was collected to obtain the raw ore material.

b. Sample 2:

We measured 10 mol/L acetic acid into a beaker, then weighed 5g minerals, stirring at room temperature under the action of a magnetic stirrer to form a uniform suspension. Next we added the mixed material to a centrifuge tube for shaking for 2 h, then let it stand. After 24h, the sample was washed with water, filtered, and finally the sample was dried in an oven at 80 °C. After the water evaporated, the minerals on the filter paper were collected to obtain an acetic acid-treated material.

c. Sample 3:

We measured 2 mol/L HF acid and placed it in a beaker, then weighed 5g minerals, stirring at room temperature under the action of a magnetic stirrer to form a uniform suspension. Next we added the mixed material to a centrifuge tube for shaking for 2 h, then let it stand. After 48h, the sample was washed with water, filtered, and placed in an oven at 80 °C to dry. After the water evaporated, we collected the minerals on the filter paper to obtain the HF acid treatment material. 

d. Sample 4:

We weighed out 1.2g Na_2_CO_3_ and 0.4g NaOH in a mortar, then weighed 5g minerals, ground in a mortar to form a uniform solid, added the uniformly mixed materials to the crucible, and roasted them at 500 °C for 2 h. The sample was then washed with deionized water, filtered, and placed in an oven at 80 °C to dry. After the water evaporated, the minerals on the filter paper were collected to obtain an alkali-treated material.

e. Sample 5:

We repeated steps b → c → d.

### 2.4. Catalyst Activity Test Method

The experimental instruments included a quartz tube, riser furnace, sampler, Fourier infrared spectrum flue gas analyzer, and computer data acquisition system. The riser furnace with a rated temperature of 1600 °C was produced by Nanjing Boyuntong Instrument Technology (Nanjing, China). A 1800 model silicon molybdenum rod with an inner diameter of 20 mm and a length of 1.2 m was heated. The FIS gas analyzer (GASMET-DX4000 model, Wuhan, China) and data acquisition system (GASMET-DX4000 model, Wuhan, China) were used for online measurement of smoke components. A reaction bed was used to support the catalyst. The experimental principle of its activity is:4NO + 4NH_3_ + O_2_ = 4N_2_ + 6H_2_O(1)
2NO_2_ + 4NH_3_ + O_2_ = 3N_2_ + 6H_2_O(2)
6NO + 4NH_3_ = 5N_2_ + 6H_2_O(3)
6NO_2_ + 8NH_3_ = 7N_2_ + 12H_2_O.(4)

Approximately 1 g of sample and 0.5g quartz wool were weighed in the heating section of the quartz tube, which is equivalent to the reaction bed used to support the catalyst. Before the start of the experiment, the vertical tube furnace was heated from room temperature to the experimental temperature at a rate of 10 °C/min. The composition of the simulated flue gas was 0.05% NH_3_, 0.05% NO, 3% O_2_; N_2_ was the equilibrium gas, and the space velocity was about 6000 h^−1^. The flow rate of the simulated flue gas was 0.1 L/min. The reaction gas was passed through for 30 min and monitored with a flue gas analyzer. After the test temperature and gas concentration were stable, the test was performed quickly. The sample was poured into a quartz tube with a constant temperature zone, and the catalyst denitration efficiency was calculated using a Fourier infrared spectrum flue gas analyzer and a computer acquisition data system.
(5)$$\eta =\frac{(NO{)}_{\mathrm{in}}-(NO{)}_{\mathrm{out}}}{(NO{)}_{\mathrm{out}}}\times 100\%,$$
where *η* is the NO removal rate, (*NO*)_in_ is the percentage of flue gas detected at the inlet of the NO, and (*NO*)_out_ is the percentage of flue gas detected at the outlet of the NO.

## 3. Results and Discussion

### 3.1. Morphological Characterization

#### 3.1.1. Scanning Electron Microscopy (SEM) Characterization

The effect of each treatment on the surface morphology of each active powder was observed by SEM. The changes of the surface morphology of the active powder, including the distribution of elements and the change of mineral phase, were analyzed.

As shown in the SEM image, the mineral material underwent different changes on the surface of the rare earth concentrate after different treatments. After treatment with acetic acid (Figure 1c) and hydrofluoric acid (Figure 1b), CaCO_3_, Ca(OH)_2_, and SiO_2_ dissolved, causing the surface of the mineral material to change from smooth to rough. After hydrofluoric acid treatment (Figure 1b), the surface of the sample became rough, and some minerals fell off to form finely divided particles. The surface showed obvious signs of acid erosion, indicating that hydrofluoric acid (Figure 1b) impregnation can improve the mineral surface area. The surface of the sample treated with acetic acid (Figure 1c) was rough, and some mineral particles were interlaced with stick-like minerals on the surface, forming a network structure, which greatly increased the surface area of the mineral. The minerals calcined by alkali (Figure 1d) had a rough interface, in which the mineral particles were partially cracked because some of the mineral phases decomposed during the calcination process, causing cracks on the surface of the minerals. The surface of the mineral calcined by acid–base (Figure 1e) was rough, and part of the crack occurred at the same time. The surface of the particles formed a flocculent package, which was converted from large to small granular minerals. In addition, the specific surface area and active sites increased. In terms of the surface morphology of the mineral material, the sample treated with acid and alkali had a larger specific surface area relative to the original ore, facilitating the adsorption and desorption of the reaction gas on the mineral surface and providing a sufficient reaction site for the occurrence of the catalytic experiment. This promoted the improvement of the denitration efficiency of the catalyst.

#### 3.1.2. Analysis of Element Types and Contents

The composition of elements in the active powder after acid and alkali treatment changed simultaneously, including the elements eluted by acid leaching and those washed out by the alkali roasting process (ass% = Element content/Total element content). In order to keep the metal elements and nonmetal elements consistent, we converted metal oxides into elemental forms.

As seen from the Table 1, phosphate, calcium salt, iron salt, bastnäsite, and a small amount of SiO_2_ were present in the ore. The calcium salt in the naturally occurring mineral material was dissolved with acetic acid, forming acid sites on the surface of the material. The calcium content of the acetic acid-treated active powder decreased by 15%. Natural minerals contain a small amount of SiO_2_. Hydrofluoric acid was added primarily to create a large specific surface area on the surface of the mineral. The introduction of hydrofluoric acid will increase the content of F. Alkali calcination decomposed the bastnäsite in the ore and removed F by converting it into NaF. The relative content of the active component effectively increased during treatment, thereby facilitating the occurrence of the catalytic reaction.

#### 3.1.3. Brunauer–Emmett–Teller (BET) Characterization

As seen from the Table 2, the specific surface area of rare earth concentrates will increase after different treatments. Alkali roasting also increases the specific surface area mainly due to the decomposition of minerals and the formation of cracks. The reason why acid treatment increases the specific surface area is that the surface is eroded and voids and depressions are formed on the surface. The acid–base-treated sample was calcined by alkali on the basis of acid–base erosion. Erosion formed on the surface, and the mineral was decomposed, forming cracks. Thus, many active components were exposed on the mineral surface. Combined with X-ray diffraction to reach a conclusion after the rare earth concentrate was treated, the specific surface area increased to 17.1 m^2^/g, exposing active substances in minerals to the surface of minerals. This will facilitate the full contact of the reaction gas with the surface adsorption site of the catalyst, increase the adsorption amount, and promote the SCR reaction. The pore volume was increased to 1.2 (mL/g), and the average pore size was reduced, forming a larger pore volume and a richer mesoporous structure. This provides more active adsorption sites for the reaction gas and facilitates the desorption and discharge of reaction products in the pores, thereby facilitating the SCR reaction.

### 3.2. Structural Characterization

#### 3.2.1. Thermogravimetric Analysis of Rare Earth Concentrates

Although the Baotou rare earth concentrate has a high rare earth content, it has more non-rare earth impurities and more complex components. To better explore the mineral phase changes of rare earth concentrates at high temperatures, a thermogravimetric analysis was conducted.

The rare earth concentrate was dried at 110 °C before the experiment to eliminate moisture. The sample was pretreated before the Thermogravimetric Analysis-Differential Scanning Calorimetry test, and the gas adsorbed by the rare earth concentrate was substantially desorbed. Before 350 °C, no significant change was observed in the TG curve, indicating that the rare earth concentrate had no weight loss and was in an exothermic state at this temperature. In the temperature range of 350–500 °C, 7% weight loss and a large peak were observed. The weight loss was due to the decomposition of CO_2_ by REFCO_3_ during roasting. The TG curve at 500–530 °C did not show weight loss, but a large endothermic peak appeared, which was caused by further oxidation of the mineral CeOF to Ce_7_O_12_. The weight loss of the TG curve at 530-650℃ is due to the conversion of Ce_7_O_12_ to CeO_2_ in the rare earth concentrate. Significant weight loss occurred in the temperature range of 650–1000 °C, and a broad endothermic peak appeared. This was due to the calcination of carbonates, and some components were sintered in rare earth concentrates. Through the TG-DSC curve, the changes of the ore phase of rare earth concentrates at different temperature ranges were observed. Fluorocarbon lanthanum was the dominant ore phase in the rare earth concentrate at different calcination temperature sections, thus providing theoretical support for the roasting test.

#### 3.2.2. X-Ray Diffraction Analysis

The Ce content in minerals treated with different concentrations of acetic acid and hydrofluoric acid did not decrease; however, the crystallinity and dispersion of minerals changed remarkably. The samples with better catalytic activity were tested as shown in Figure 2.

As shown in the figure, acetic acid and hydrofluoric acid treatment did not lead to a new mineral phase compared with the original ore. In addition, the original ore phase did not disappear; however, the diffraction peak of some mineral phases became sharper. This finding indicated that, during the acid impregnation process, the crystal structure of the ore changed, and the Ce that was encapsulated and embedded in the mineral was exposed to the surface of the mineral, increasing the crystallinity of the crystal. Acetic acid and hydrofluoric acid impregnation effectively increased the content of rare earth compounds on the surface of rare earth concentrates, whereas acid erosion increased the specific surface area of rare earth concentrates, providing more acid sites for gases in the catalytic process and improving the catalytic performance of the active powder.

The difference between the diffraction peaks of the active powder calcined at 500 °C and the original ore after co-treatment with acetic acid, hydrofluoric acid, and alkali treatment was great. Compared with CePO_4_ and Ca(PO_4_)_3_F in the ore phase, the treated active powder partially decomposed rare earth elements while retaining the original ore phase and forming Ce_7_O_12_ and transition metal oxide (such as Fe_2_O_3_). This caused the rare earth element and the transition metal element to form a composite oxide and promoted the catalytic reaction efficiency of the active powder. The co-treatment of acetic acid and hydrofluoric acid not only improved the defluorination efficiency, but also reduced impurities while increasing the CeO_2_ content and optimizing the dispersion of CeO_2_, thereby effectively improving the activity of the catalyst.

#### 3.2.3. X-Ray Photoelectron Spectroscopy (XPS) Analysis

According to reports in the literature, there are two forms of Ce present in minerals. As shown in the Figure 3, one is Ce^3+^: u_0_ (BE ≈ 884.4 eV) and u_0_’ (BE ≈ 903.9 eV); the second is Ce^4+^: v_0_ (BE ≈ 882.2 eV), v_1_ (BE ≈ 888.6 eV), v_2_ (BE ≈ 898 eV), V_0_’ (BE ≈ 900.7 eV), v_1_’ (BE ≈ 907.2 eV), and v_2_’ (BE ≈ 916.15 eV) [20,21]. Ce in sample 1 was converted from Ce^3+^/Ce^4+^+Ce^3+^ = 21.61% to Ce^3+^/Ce^4+^+Ce^3+^ = 61.73% in sample 5. Therefore, the greater the content of Ce^3+^, the better the denitration activity of the catalysts in the low-temperature window.
Ce^3+^ + O_2_ + v_0_→Ce^4+^ + O_2_^−^(6)

The Ce^3+^ content of the ore was quantitatively analyzed by XPS. The Ce^4+^ of the acid–base-treated active powder had a new peak in v_1_, v_2_, and v_2_’ relative to the original ore, and the corresponding peak areas of v_0_, v_0_’, and v_1_’ correspondingly increased. This finding indicated that the content of Ce^4+^ in the acid–base-treated active powder was increased relative to the ore. The acid–base treatment was beneficial to the conversion of Ce^3+^ to Ce^4+^. The valence state of Ce existed simultaneously with +4 and +3 [22]. This facilitates the storage and release of surface oxygen, thereby increasing surface oxidation and promoting the reduction of nitrogen oxides absorbed on the surface of the active powder.

According to the literature, there are two forms of Fe present in minerals. One is Fe^2+^: h_0_ (BE ≈ 709.8 eV), h_0_’ (BE ≈ 722.8 eV), h_1_ (BE ≈ 716.4 eV), and h_1_’ (BE ≈ 730.0 eV); the second is Fe^3+^: I_0_ (BE ≈ 711.2 eV), I_0_’ (BE ≈ 723.4 eV), I_1_ (BE ≈ 719.5 eV), and I_1_’ (BE ≈ 733.6 eV) [23]. As shown in the figure, Fe in sample 1 was transformed from Fe^2+^/Fe^2+^+Fe^3+^ = 35.17% to Fe^2+^/Fe^2+^+Fe^3+^ = 53.35% in sample 5. Therefore, the greater the Fe^2+^ content, the more the oxygen hole content in the catalyst and the better the low-temperature denitration activity of the catalyst.
Fe^2+^ + O_2_ + v_0_→Fe^3+^ + O_2_^−^(7)

According to the semi-quantitative relationship of the fitted peak areas, the active powders subjected to acid–base co-treatment had relatively higher Fe^3+^ content than the ore. The transition metal element Fe can provide electrons required for the reaction during the SCR reaction, thereby promoting the catalytic reaction of the active powder [24]. The formation of Fe^2+^ and Ce^3+^ follows the conservation of charge. When substances with different valence states are formed, oxygen vacancies and adsorption sites are formed on the surface of the mineral catalyst. At the low-temperature stage, NH_3_ adsorbed on the surface of the mineral catalyst reacts with oxygen adsorbed on the surface to form NH_2_ and ‒OH. As the temperature increases, NH_2_ will react with NO adsorbed on the surface to form NH_2_NO intermediates. Under certain conditions, NH_2_NO decomposes to form N_2_ and H_2_O. Therefore, the increase of oxygen vacancies and adsorption sites on the catalyst surface through loading has a significant influence on the catalytic efficiency of the catalyst [25,26].

#### 3.2.4. H_2_-Temperature-Programmed Reduction (H_2_-TPR) Analysis

To investigate the ability of metal ions on the surface of modified rare earth concentrates to be reduced to low-valence metal ions and to absorb or release oxygen, an H_2_-TPR experiment was carried out. The results are shown in Figure 4.

As shown in the figure, the rare earth concentrate plus alkali-calcined active powder and the acetic acid-impregnated active powder showed a weak peak at 400 °C. At 500–550 °C, the alkali-calcined active powder, acetic acid-impregnated active powder, hydrofluoric acid-impregnated active powder, and ore showed strong peaks, while the acid‒base co-processed active powder was in the range of 500–550 °C. A strong spike and shoulder were present at 750 °C. In addition, a small shoulder appeared in the ore, acetic acid-impregnated active powder, and hydrofluoric acid-impregnated active powder at 600–700 °C. The rare-earth mineral had a de-oxidation peak between 500 °C and 550 °C. The reason was that Fe_2_O_3_ was converted into Fe_3_O_4_ after being combined with Ce in rare earth concentrate. Among them, the de-oxidation peak between 500 °C and 600 °C corresponded to the conversion process of Fe_3_O_4_→FeO→Fe. The de-oxidation peak between 600 °C and 750 °C was attributable to the synergistic effect between Fe and rare earth concentrate. Ce^4+^ was converted into Ce^3+^, namely, CeO_2_ was converted into Ce_2_O_3_. The alkali-calcined active powder had a broad peak, and its peak area was 547.62. The area of the acid–base co-processed active powder was 641.74, which was 17.19% higher than that of alkali treatment [27]. The peak area of the active powder after acid–base co-treatment was greatly improved compared with that of the ore. The active powder prepared by acid leaching and alkali roasting exhibited a good redox ability.

#### 3.2.5. NH_3_-Temperature-Programmed Desorption (NH_3_-TPD) and NO-TPD Analysis

To investigate whether the modification treatment affected the surface acidity of the rare earth concentrate, a NH_3_-TPD experiment was carried out on the rare earth concentrates with different modification treatments. The samples were degassed with Ar gas at 20 L/min for 30 min before running NH_3_-TPD and H_2_-TPR. The test results are shown in Figure 5.

As shown in the figure, the surface acid amount (from higher to lower) was ordered as follows: acid–base treatment > acetic acid treatment > hydrofluoric acid treatment > additional alkali roasting treatment. Four samples had a weak desorption peak at the low-temperature range (100–250 °C). The peak corresponded to the absorption of NH_3_ on the weak acid site. A continuous, wide, and strong NH_3_ desorption peak appeared at the high-temperature range (250–900 °C), which was absorbed by the strong acid site. It was produced after NH_3_ desorption. This shows that a large number of substances work together to increase the number of acidic sites and surface active sites on the surface of the material, resulting in a large amount of NH_3_ adsorption on the surface of the material. This finding shows that the surface of particles can be modified to make it rough and porous after the acid–base roasting treatment.

NO can participate in the SCR reaction in the form of adsorption. Therefore, the adsorption capacity of NO can affect the SCR reactivity to some extent. To determine the effect of mineral modification on the adsorption capacity of NO, this paper does NO for the four catalysts. The results of the temperature-programmed adsorption–desorption test are shown in Figure 6.

As shown in the figure, the desorption process of the ore occurred between 400 °C and 900 °C. The desorption curve after acid treatment occurred at 400–600 °C, and the area of the original ore peak was significantly increased. The desorption curve of the concentrate after alkali treatment was between 600 °C and 700 °C, and the area of the desorption peak was also significantly increased compared with that of the original ore. The NO adsorption effect of the rare earth concentrate after alkali and acid treatment was obviously improved. The NO-TPD peak shape after alkali treatment was stronger than that of the ore, and the desorption peak was enhanced and shifted to the high-temperature direction. The peak value of NO-TPD did not shift significantly after acid treatment; however, the peak width and peak height increased significantly. The acid and alkali treatments had obvious promotion effects on the NO adsorption capacity of rare earth concentrates. This indicates that a large number of substances worked together to absorb NO on the surface of the catalyst, increasing the surface acid amount and surface active sites. The acid–base treatment could modify the surface of the particle to make it rough and porous.

### 3.3. SCR Activity of Catalyst

The NO conversion rate of the samples was measured using a simulated flue gas device. In the best state, the NO conversion rate of the rare earth concentrate powder is only 36.9%. Acetic acid dissolves CaCO_3_ in the rare earth concentrate and forms a large specific surface area on the surface of the mineral. At the same time, more elements such as CeFe in the mineral are exposed. The NO conversion efficiency of reached 83.5% at 350 °C. The HF acid destroyed SiO_2_ in the rare earth mineral, collapsed the natural skeleton in the mineral, changed the surface pore structure of the mineral from macropores to micropores, further improved the specific surface area of the rare earth concentrate, and promoted the catalytic efficiency of the material. The NO conversion efficiency of Sample 3 reached 78.9% at 450 °C. High-temperature roasting of rare earth concentrates and mixed alkalis promoted the conversion of Ce, Fe, and other substances into oxides and the formation of cracks due to mineral instability under high-temperature conditions. Sample 4: the catalytic activity reached 69.9% at 450 °C. The specific surface area of the sample treated with acid and alkali was greatly increased; at the same time, oxides of elements such as Ce and Fe formed on the mineral surface, and a layer of highly efficient catalytic material formed on the mineral surface. Sample 5: The catalytic activity reached 91.3% at 450 °C. The NO activity reached 87.4% at 400 °C because SiO_2_, CaF, and other calcium salts were partially dissolved, increasing the specific surface area, which is beneficial to catalytic activity. Calcination caused the effective elements to become exposed on the surface of the mineral material to form a solid solution, thereby improving the catalytic activity.

## 4. Conclusions

The surface of the catalyst was uneven, the specific surface area increased, the average pore diameter decreased, and the pore volume increased. In the active component of the catalyst, Ce coexisted in the forms of Ce^3+^ and Ce^4+^, whereas Fe coexisted in the forms of Fe^3+^ and Fe^2+^. Moreover, the content of Fe^3+^ was higher than that of Fe^2+^. The acid–base co-processed sample decomposed rare earth elements and transition metal elements by retaining the ore phase to form a certain amount of Ce_7_O_12_ and transition metal oxide (such as Fe_2_O_3_). Acetic acid dissolved the calcium salt in the naturally occurring mineral material, increasing the specific surface area of the catalyst and forming several acid sites on the surface of the material. In general, the relative content of the active components of the acid–base-treated samples increased, which promoted the conversion of NO, thereby facilitating the SCR reaction. The NO conversion of the sample was measured using a simulated flue gas device. In the best state, the NO conversion rate of the untreated rare earth concentrate powder was only 36.9%. The acid–base-treated rare earth concentrates had a denitration efficiency of 87.4% at a reaction temperature of 400 °C. 

## Figures and Tables

**Figure 1 materials-13-00580-f001:**
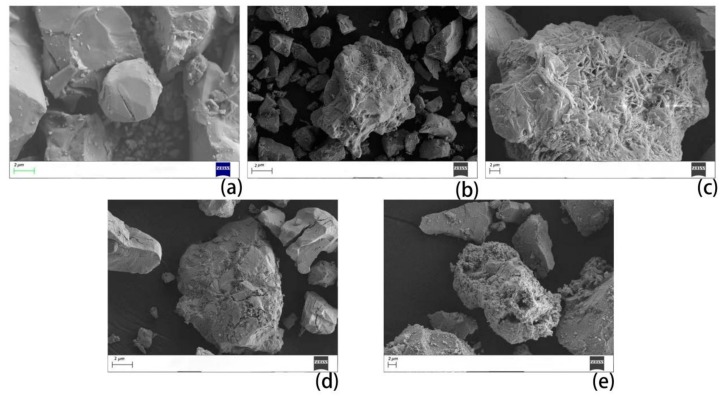
SEM characterization of rare earth concentrates subjected to different treatments. (**a**) Sample 1, (**b**) Sample 2, (**c**) Sample 3, (**d**) Sample 4, (**e**) Sample 5.

**Figure 2 materials-13-00580-f002:**
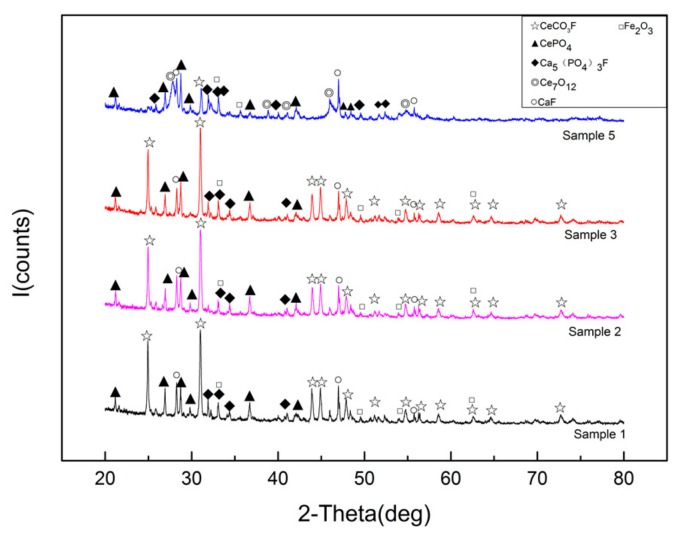
Analysis of rare earth concentrate ore phase subjected to different treatments.

**Figure 3 materials-13-00580-f003:**
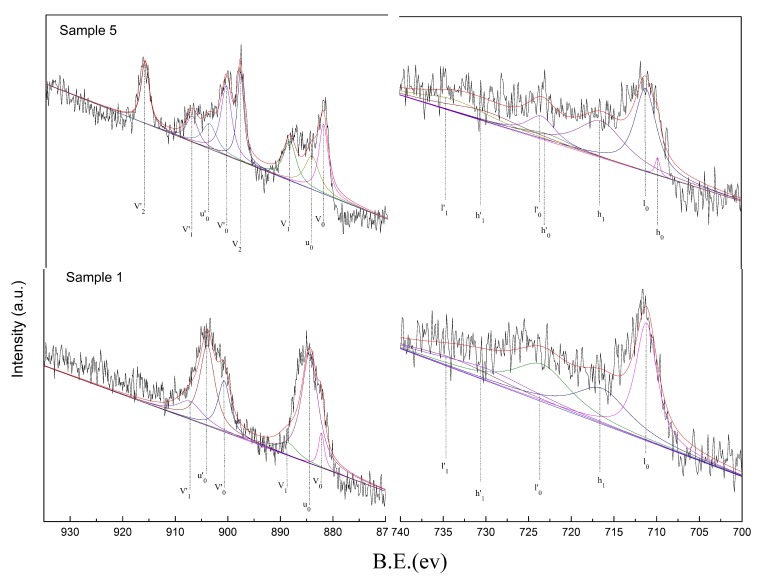
XPS energy spectrum of Ce and Fe 2p orbital on the surface of ore and acid‒base co-processed active powder.

**Figure 4 materials-13-00580-f004:**
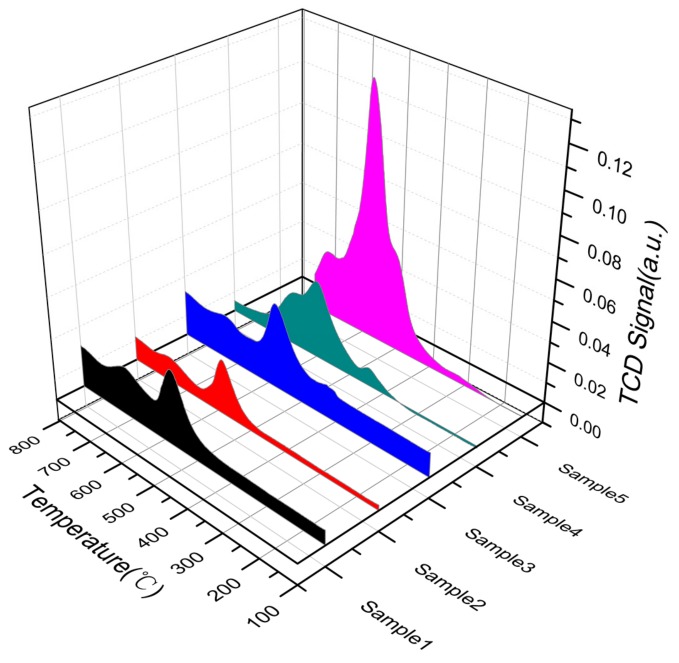
Analysis of the redox capability of rare earth concentrates subjected to different treatments.

**Figure 5 materials-13-00580-f005:**
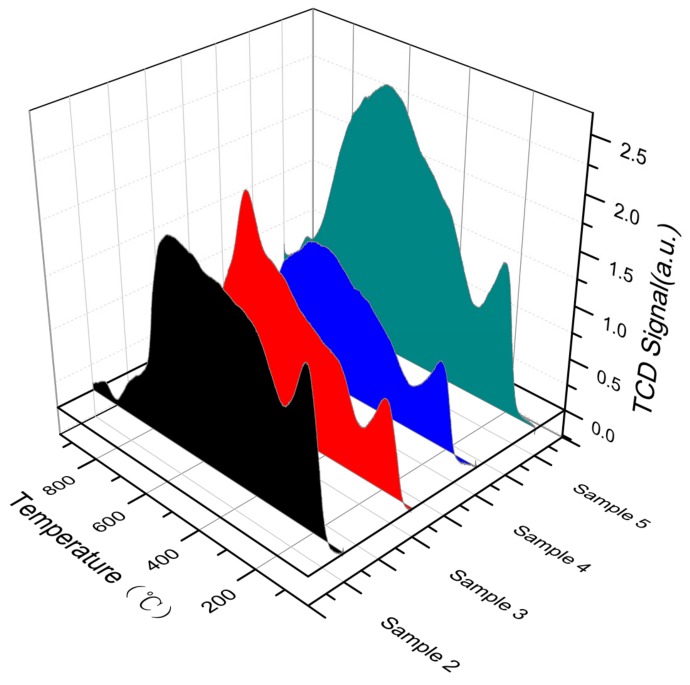
Characterization of NH_3_-TPD in rare earth concentrates subjected to different treatments.

**Figure 6 materials-13-00580-f006:**
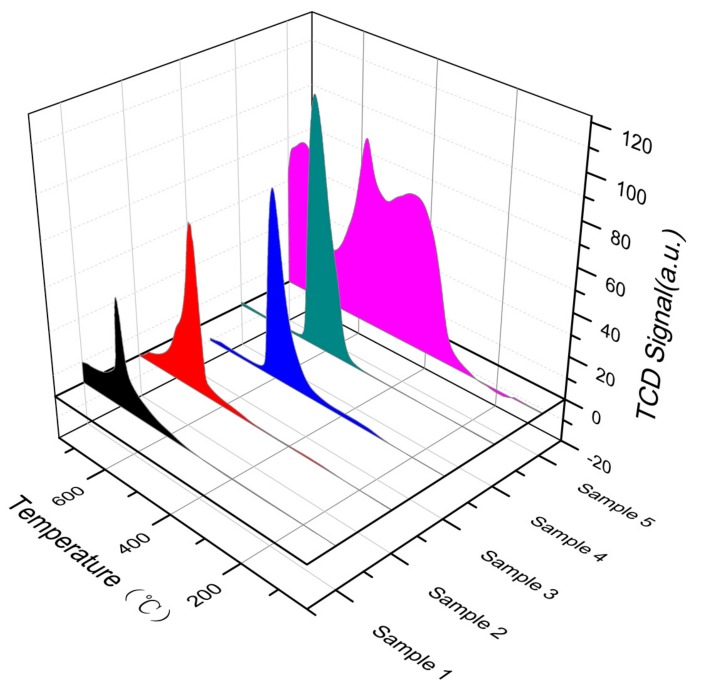
Adsorption characteristics of NO in samples subjected to different treatment methods.

**Table 1 materials-13-00580-t001:** XRF characterization of rare earth concentrates.

Element	Sample 1 (%)	Sample 2 (%)	Sample 3 (%)	Sample 4 (%)	Sample 5 (%)
	(Raw Ore)				
F	6	5	6	3	3
Na	≤1	1	≤1	≤1	≤1
Mg	≤1	≤1	≤1	≤1	0.4
Al	≤1	≤1	≤1	≤1	≤1
Si	3	2	1	2	≤1
P	12	7	3	10	1
S	5	5	3	4	1.1
K	≤1	≤1	≤1	≤1	≤1
Ca	20	8	15	19	4
Ti	≤1	≤1	≤1	≤1	≤1
Fe	11	10	10	8	12
Zn	2	≤1	≤1	≤1	≤1
Sr	≤1	≤1	≤1	≤1	≤1
Nb	≤1	≤1	≤1	≤1	≤1
Ba	≤1	≤1	≤1	≤1	≤1
La	10	9	13	12	18
Ce	16	33	32	29	40
Pr	3	5	5	4	8
Nd	9	8	5	5	7
Pb	2	6	5	2	3
Mn	≤1	≤1	≤1	≤1	1
Pd	≤1	≤1	≤1	≤1	≤1
Th	≤1	≤1	≤1	≤1	≤1

**Table 2 materials-13-00580-t002:** BET characterization of rare earth concentrates under different conditions.

	Specific Surface Area (m^2^/g)	Average Aperture (nm)
Sample1	0.7	1.24
Sample2	7.8	6.86
Sample4	8.3	1.15
Sample3	8.5	2.98
Sample5	17.1	1.19

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
