# Peer review of "Surface Properties and Denitrification Performance of Impurity-Removed Rare Earth Concentrate"

_materials, 2020, doi:10.3390/ma13030580_

Round 1
Reviewer 1 Report
The manuscript presents the studies of modified rare earth concentrate as catalysts for NH3-SCR process. The quality of the presented studies is weak and should be significantly improved.
The selectivity to the reaction products is not presented and disuses. The studies of NH3-SCR without analysis of the reaction selectivity cannot be accepted. This issue has to be presented and discussed in the revised version of the manuscript; The profiles of NO conversion has to be presented and discussed in the revised version of the manuscript; The composition of the reaction mixture used in the catalytic studies should be included in the revised version of the manuscript; Were the samples outgassed prior to the NH3-TPD and H2-TPR runs? Discussion of the H2-TPR results limited only to the identification of the reduction peaks maxima in pointless. The samples contain various components that can be reduced in the temperature range of H2-TPR experiment. Reduction peaks should be related to the reduction of the specific components of the samples; There is pointless to discuss the nature of the acid sites, Brønsted and Lewis, based only on the NH3-TPD results. If authors really would like to analyses and discuss this problem additional experiments should be done (e.g. FT-IR analysis of the samples pre-adsorbed with pyridine); The correlation between physicochemical properties of the samples and their catalytic activity should be presented and discussed.Author Response
Dear Editors and Reviewers:
Thank you for your letter and for the reviewers’ comments concerning our manuscript entitled “Surface Properties and Denitrification Performance of Impurity-Removed Rare Earth Concentrate” (ID: materials-688919).Those comments are all valuable and very helpful for revising and improving our paper, as well as the important guiding significance to our researches. We have studied comments carefully and have made correction which we hope meet with approval. Revised portion are marked in red in the paper. The main corrections in the paper and the responds to the reviewer’s comments are as flowing:
Responds to the reviewer’s comments:
Reviewer :
Response to comment:
The manuscript presents the studies of modified rare earth concentrate as catalysts for NH3-SCR process. The quality of the presented studies is weak and should be significantly improved.The selectivity to the reaction products is not presented and disuses. The studies of NH3-SCR without analysis of the reaction selectivity cannot be accepted. This issue has to be presented and discussed in the revised version of the manuscript.
Response:
The manuscript discusses the study of selective catalytic reduction of NH3-SCR, and the amendments are as follows:
A reaction bed was used to support the catalyst. The experimental principle of its activity is:
4NO + 4NH3 + O2 = 4N2 + 6H2O |
(1) |
2NO2 + 4NH3 + O2 = 3N2 + 6H2O |
(2) |
6NO + 4NH3 = 5N2 + 6H2O |
(3) |
6NO2 + 8NH3 = 7N2 + 12H2O |
(4) |
Therefore, the greater the content of Ce3+, the better the denitration activity of the catalysts in the low-temperature window.
Ce3++O2+v0→Ce4++O2- (6)
Therefore, the greater the Fe2+ content, the more the oxygen hole content in the catalyst and the better the low-temperature denitration activity of the catalyst.
Fe2++O2+v0→Fe3++O2- (7)
When substances with different valence states are formed, oxygen vacancies and adsorption sites are formed on the surface of the mineral catalyst. At the low temperature stage, NH3 adsorbed on the surface of the mineral catalyst reacts with oxygen adsorbed on the surface to form NH2 and -OH. As the temperature increases, NH2 will react with NO adsorbed on the surface to form NH2NO intermediates. Under certain conditions, NH2NO decomposes to form N2 and H2O. Therefore, the increase of oxygen vacancies and adsorption sites on the catalyst surface through loading has a great influence on the catalytic efficiency of the catalyst.
The profiles of NO conversion has to be presented and discussed in the revised version of the manuscript; The composition of the reaction mixture used in the catalytic studies should be included in the revised version of the manuscript.
Response:
Relevant experimental conditions for catalytic tests have been supplemented.and the amendments are as follows:
The composition of the simulated flue gas is 0.05% NH3, 0.05% NO, 3% O2, and N2 is the equilibrium gas, and the space velocity is about 6000 h-1. The flow rate of the simulated flue gas is 0.1 L/min. The reaction gas was passed through for 30 min and monitored with a flue gas analyzer.
Were the samples outgassed prior to the NH3-TPD and H2-TPR runs?
Response:
Yes.and the amendments are as follows:
And the samples were outgassed prior to the NH3-TPD and H2-TPR runs.
The samples contain various components that can be reduced in the temperature range of H2-TPR experiment.Reduction peaks should be related to the reduction of the specific components of the samples; Discussion of the H2-TPR results limited only to the identification of the reduction peaks maxima in pointless.
Response:
We have re-written this part according to the Reviewer’s suggestion. The Rare-earth mineral had a de-oxidation peak between 500 ℃~550 ℃. The reason was that Fe2O3 was converted into Fe3O4 after being combined with Ce in rare earth concentrate. Among them, the de-oxidation peak between 500℃~600℃corresponded to the conversion process of Fe3O4→FeO→Fe. The de-oxidation peak between 600 ℃~750 ℃ was attributable to the synergistic effect between Fe and rare earth concentrate, Ce4+ was converted into Ce3+, namely CeO2 was converted into Ce2O3.
There is pointless to discuss the nature of the acid sites, Brønsted and Lewis, based only on the NH3-TPD results. If authors really would like to analyses and discuss this problem additional experiments should be done (e.g. FT-IR analysis of the samples pre-adsorbed with pyridine); The correlation between physicochemical properties of the samples and their catalytic activity should be presented and discussed.
Response:
It is really true as Reviewer suggested that due to the limitation on the number of figures and tables in the manuscript, we deleted the relevant data of FT-IR analysis. We focus our research on the performance of mineral catalytic materials prepared by different preparation methods. The FT-IR analysis of rare earth concentrates will be further studied in the future.
Reviewer 2 Report
It is hardly/impossible to see "the advance the in-depth understanding of the relationship between the structure, the properties or the functions of all kinds of materials." according to the aim of the journal.The manuscript is more like a technical note and not a research study.
In my opinion the manuscript is more appropriate for geology/mineralogy related journals and I believe that it will accepted.
Author Response
Dear Editors and Reviewers:
Thank you for your letter and for the reviewers’ comments concerning our manuscript entitled “Surface Properties and Denitrification Performance of Impurity-Removed Rare Earth Concentrate” (ID: materials-688919).Those comments are all valuable and very helpful for revising and improving our paper, as well as the important guiding significance to our researches. We have studied comments carefully and have made correction which we hope meet with approval. Revised portion are marked in red in the paper. The main corrections in the paper and the responds to the reviewer’s comments are as flowing:
Responds to the reviewer’s comments:
It is hardly/impossible to see "the advance the in-depth understanding of the relationship between the structure, the properties or the functions of all kinds of materials." according to the aim of the journal.The manuscript is more like a technical note and not a research study.In my opinion the manuscript is more appropriate for geology/mineralogy related journals and I believe that it will accepted.
Response:
We greatly appreciate the reviewers' recognition of our work. The traditional catalyst preparation process has high preparation cost and complicated process. At present, the research of green catalysts based on mineral catalysts has become a hot spot. The content of the manuscript describes the preparation process of the catalyst material, while paying more attention to the characterization of the catalyst material and the verification of the material properties. So we think it is more suitable to publish on Materials.
We tried our best to improve the manuscript and made some changes in the manuscript. These changes will not influence the content and framework of the paper, And marked in red in revised paper.
We appreciate for Editors and Reviewers’ warm work earnestly, and hope that the correction will meet with approval.
Once again, thank you very much for your comments and suggestions.
Thank you and best regards.
Yours sincerely,
Wen-fei Wu
Corresponding author: Wen-fei Wu
Name: Wen-fei Wu
E-mail: [email protected] (Prof. Wen-fei Wu)

Reviewer 3 Report
Interesting and creative work!
Big Problem: In Table 1 we find that the concentration of Ce doubled with the various treatments. That implies that lots of other elements had to be removed. But if you look at the other elements, not enough of them were removed by the processing. You need to remove about 50% of the other elements to double the Ce. Not the case. That invalidates the entire paper.
Also, this table seems impossible because there are way, way too few anions present to balance the cations. Were most of the cations present as metal?
(Maybe you did not grind up the material before you put it in the XRF, as is the correct way to perform XRF analysis. If so, material at the surface of the sample will be better sampled by the XRF that that deeper in the sample. But then you really don't know what you are analyzing.)
Conclusions should point out which process would be best from an economic standpoint - % improvement versus cost of process.
Similarly, does this process make any economic sense: use the concentrate to make low-value Ce (16%)-based catalyst and waste (leave in the catalyst) the high-value didymium (11%)? I would always choose to go after the neodymium and praseodymium.
General comment - way too many significant figures are given, e.g., 5 in some places in Table 1. XRF has no such precision.
Author Response
Thank you for your letter and for the reviewers’ comments concerning our manuscript entitled “Surface Properties and Denitrification Performance of Impurity-Removed Rare Earth Concentrate” (ID: materials-688919).Those comments are all valuable and very helpful for revising and improving our paper, as well as the important guiding significance to our researches. We have studied comments carefully and have made correction which we hope meet with approval. Revised portion are marked in red in the paper. The main corrections in the paper and the responds to the reviewer’s comments are as flowing:
Responds to the reviewer’s comments:
It is hardly/impossible to see "the advance the in-depth understanding of the relationship between the structure, the properties or the functions of all kinds of materials." according to the aim of the journal.The manuscript is more like a technical note and not a research study.In my opinion the manuscript is more appropriate for geology/mineralogy related journals and I believe that it will accepted.
3.Response to comment:
Big Problem: In Table 1 we find that the concentration of Ce doubled with the various treatments. That implies that lots of other elements had to be removed. But if you look at the other elements, not enough of them were removed by the processing. You need to remove about 50% of the other elements to double the Ce. Not the case. That invalidates the entire paper.
Response:
We have re-written this part according to the Reviewer’s suggestion. In order to make the data in the table more clearly and accurately reflect the content contained, in order to keep the metal elements and non-metal elements consistent, the metal oxides are converted in the form of elemental elements. From the perspective of element content, the increase of Ce is accompanied by the decrease of other elements, but it does not mean that when Ce increases by 50%, other elements must be reduced by 50%. Because the changes in the content of the elements are relative. Each step of the sample preparation process is a process study based on retaining Ce. We have made correction according to the Reviewer’s comments.
In order to keep the metal elements and non-metal elements consistent, we converted metal oxides into elemental forms.
This table seems impossible because there are way, way too few anions present to balance the cations. Were most of the cations present as metal?(Maybe you did not grind up the material before you put it in the XRF, as is the correct way to perform XRF analysis. If so, material at the surface of the sample will be better sampled by the XRF that that deeper in the sample. But then you really don't know what you are analyzing.)
Response:
In the XRF detection room, the sample was dissolved in the microwave digestion apparatus using concentrated HNO3 to make the minerals dissolve, and then the test was performed in the form of liquid. Because there are many types of elements in the mineral structure, there are various ways to combine the elements. Most metal ions exist as cations, but not all cations are metal ions. The changes in element content during the preparation process are relative, so the charge balance between anions and cations is not limited to metal and non-metal elements, but also exists between non-metal and non-metal elements. The law of changing the content of elements is mainly used to explore the effect of changes in the elements on the catalytic performance of mineral catalysts, such as the increase or decrease of certain elements leads to the increase in catalytic performance.
Conclusions should point out which process would be best from an economic standpoint - % improvement versus cost of process.
Response:
We have re-written this part according to the Reviewer’s suggestion.
The NO conversion of the sample was measured using a simulated flue gas device. In the best state, the NO conversion rate of the untreated rare earth concentrate powder is only 36.9%. The acid–base-treated rare earth concentrates had a denitration efficiency of 87.4% at a reaction temperature of 400 ℃.
Similarly, does this process make any economic sense: use the concentrate to make low-value Ce (16%)-based catalyst and waste (leave in the catalyst) the high-value didymium (11%)? I would always choose to go after the neodymium and praseodymium.
Response:
Compared with the traditionally prepared catalytic materials, we use minerals directly to prepare catalytic materials to save complicated purification processes and preparation processes, and use the structure of natural minerals to directly prepare mineral catalytic materials. Based on the main elements in traditional catalysts and after determining the main components in the mineral, we chose Ce and Fe as the main research objects. Because other elements were unstable during processing, no other elements were investigated. However, in the future research process, we will further study the relationship between all the elements in the mineral and the properties of the catalytic material.
General comment - way too many significant figures are given, e.g., 5 in some places in Table 1. XRF has no such precision.
Response:
We have made correction according to the Reviewer’s comments.
Element |
Sample 1(%) |
Sample 2(%) |
Sample 3(%) |
Sample 4(%) |
Sample 5(%) |
(Raw ore) |
|||||
F |
6 |
5 |
6 |
3 |
3 |
Na |
≤1 |
1 |
≤1 |
≤1 |
≤1 |
Mg |
≤1 |
≤1 |
≤1 |
≤1 |
0.4 |
Al |
≤1 |
≤1 |
≤1 |
≤1 |
≤1 |
Si |
3 |
2 |
1 |
2 |
≤1 |
P |
12 |
7 |
3 |
10 |
1 |
S |
5 |
5 |
3 |
4 |
1.1 |
K |
≤1 |
≤1 |
≤1 |
≤1 |
≤1 |
Ca |
20 |
8 |
15 |
19 |
4 |
Ti |
≤1 |
≤1 |
≤1 |
≤1 |
≤1 |
Fe |
11 |
10 |
10 |
8 |
12 |
Zn |
2 |
≤1 |
≤1 |
≤1 |
≤1 |
Sr |
≤1 |
≤1 |
≤1 |
≤1 |
≤1 |
Nb |
≤1 |
≤1 |
≤1 |
≤1 |
≤1 |
Ba |
≤1 |
≤1 |
≤1 |
≤1 |
≤1 |
La |
10 |
9 |
13 |
12 |
18 |
Ce |
16 |
33 |
32 |
29 |
40 |
Pr |
3 |
5 |
5 |
4 |
8 |
Nd |
9 |
8 |
5 |
5 |
7 |
Pb |
2 |
6 |
5 |
2 |
3 |
Mn |
≤1 |
≤1 |
≤1 |
≤1 |
1 |
Pd |
≤1 |
≤1 |
≤1 |
≤1 |
≤1 |
Th |
≤1 |
≤1 |
≤1 |
≤1 |
≤1 |
We tried our best to improve the manuscript and made some changes in the manuscript. These changes will not influence the content and framework of the paper, And marked in red in revised paper.
We appreciate for Editors and Reviewers’ warm work earnestly, and hope that the correction will meet with approval.
Once again, thank you very much for your comments and suggestions.
Thank you and best regards.
Yours sincerely,
Wen-fei Wu
Corresponding author: Wen-fei Wu
Name: Wen-fei Wu
E-mail: [email protected] (Prof. Wen-fei Wu)

Round 2
Reviewer 1 Report
Authors still did not included any results presenting selectivity of the reaction conducted in the presence of the studied catalysts. As I mentioned in the previous review rapport such information is necessary for the proper catalyst evaluation. Are Authors sure that only dinitrogen was produced, what about formation of N2O?
Moreover, authors stated that the catalysts were outgassed prior to the TPD and TPR runs. What was the conditions of the outgassing?
I cannot accept the manuscript in the present form.
Author Response
Authors still did not included any results presenting selectivity of the reaction conducted in the presence of the studied catalysts. As I mentioned in the previous review rapport such information is necessary for the proper catalyst evaluation. Are Authors sure that only dinitrogen was produced, what about formation of N2O?
Response:We have done the corresponding work and have made some progress. However, the research content of this article is mainly on the process of preparing rare earth mineral catalysts from rare earth concentrates, and Due to the limitations of article graphics and tables, we additionally make the following response to your thoughts. Hope to get your approval.
NH3adsorption
Fig. 1.Adsorption of NH3 on the surface of catalyst at different temperatures.
The in situ DRIFT spectrum was tested and the results are shown in Fig. 1. Prior to the gas adsorption experiment, sample 5 was held at 300 ℃ for 2 h with N2 to blow off CO2 and H2O in the air, and then the background spectrum was measured under the same conditions. When cooled to a target temperature of 50 ℃, 0.05% of NH3 was pumped into the system for 30 min and then the in-situ DRIFT spectrum was measured as the temperature increased. Among them, 1143 cm-1 is attributed to NH3 at the Lewis acid position, 1452 cm-1 is attributed to the symmetric deformation vibration adsorption peak of NH4+ at B-acid position, and 1317 cm-1 is attributed to collaborative vibration adsorption peak of NH3 at Lewis. As the temperature increases, an intermediate product adsorption peak of NH3 oxidation occurs at 1565 cm-1 because the amino species adsorbed on the surface of the catalyst is oxidized by a small amount of adsorbed oxygen inside the catalyst as the temperature increases. At the same time, the adsorption peak of the B-acid site gradually weakened and almost completely disappeared at around 250℃. That is the on the surface of the catalyst. At low temperature, the adsorption of L-acid sites NH3 and B-acid sites NH4+ is present on the surface of the catalyst. As the temperature increases, the adsorbed material of the Bacid site NH4+ desorbs from the surface, and most of the NH3 adsorbed at the L-acid site exists.
NO +O2 adsorption
Fig.2.Adsorption of NO + O2 on the surface of catalyst at different temperatures.
The in situ DRIFT spectrum test results are shown in Fig.2. Before the gas adsorption experiment, sample 5 was placed at 300 ℃, and CO2 and H2O in the air were blown off with N2 for 2 h, and then the background spectrum was measured under the same conditions. When cooled to the target temperature of 50 ℃, 0.05% NO+3% O2 was pumped into the system for 30 min, and then the in-situ drift spectrum was measured as the temperature increased. The in-situ DRIFT spectrum of NO+O2 adsorption is shown in Fig. 1. Several peaks of 1155 cm-1 , 1290 cm-1 , 1526 cm-1 and 1610 cm-1 appear in the range of 1000–2000 cm-1. Among them, 1155 cm-1 belongs to the adsorption peak of bridge nitrite, and 1526 cm-1 is the double-tooth nitrate adsorption peak. 1610 cm-1 is attributed to the weakly adsorbed NO2 on the catalyst surface, and 1290 cm-1 belongs to the double-toothed subAsian Nitrate adsorption peak. And it can be clearly seen that as the adsorption temperature increases, the intensity of the peak increases. Cracks appear on the surface of the treated sample, which increases the probability of contact between NOx and the catalyst. At the same time, generated Ce oxide also has a strong adsorption capacity, so that more NO changes on the surface of the catalyst, or adsorbs or oxidizes. A nitrate substance is formed.
Reactions between NO + O2and adsorbed NH3 species
Fig.3.. Adsorption spectrum of NO + O2 on the surface of cataly stadsorbed by NH3.
The in situ DRIFT spectrum test results are shown in Fig.3. Prior to the gas adsorption experiment, Sample 5 was placed at 300 ℃, and CO2 and H2O in the air were blown off with N2 for 30 min, and then the background spectrum was measured under the same conditions. When cooled to a target temperature of 150 ℃, NH3 was introduced. After the NH3 adsorption is completed, purge with N2 and record the corresponding spectrum. After the adsorption is completed, the in-situ drift spectrum is then measured as the temperature rises. NO+O2 has introduced again, and the corresponding spectrum is recorded in time. In order to investigate the reactivity of the pre-adsorbed NH3 with the NO+O2 species in the SCR reaction on the surface of sample 5, the in itu DRIFTS of the reaction between the pre-adsorbed NOx and NH3 at 200 ℃ was tested, and the results are shown in Fig.2. The surface has an adsorption peak of 1445 cm-1 attributed to the NH4+ of the B-acid site, and there are also adsorption peaks of 1151 cm-1 and 1279 cm-1 attributed to the L-acid position NH3, and there is also an intermediate attributed to the formation of NH3 oxidation. The adsorption peak at 1522 cm-1 of the substance. After the introduction of NO + O2, the adsorption peaks of NH4+ and L-acid sites NH3 attributed to B-acid sites disappeared within 10 min. The adsorption peaks attributable to the bidentate nitrate at 1082 cm-1 and 1522 cm-1 , monumental nitrate at 1279 cm-1 and the bridge nitrate at 1151 cm-1 were formed. This indicates that gaseous NOX reacts with NH3 previously adsorbed on the Lewis acid site and the Brønsted acid site. After the adsorbed NH3 is reacted, the active sites on the surface of the catalyst are occupied by the nitrate/nitrite species.
Reactions between NH3and adsorbed NO+ O2 species
Fig.4.Adsorption spectrum of NH3 on the surface of catalyst pre-adsorbed with NO+O2.
The in situ DRIFT spectrum test results are shown in Fig.4. Prior to the gas adsorption experiment, Sample 5 was placed at 300℃, CO2 and H2O in the air were blown off with N2 for 30 min, and then the background spectrum was measured under the same conditions. When cooled to a target temperature of 150 ℃, NO+ O2 was introduced. After the NO + O2 adsorption is completed, purge with N2 and record the corresponding spectrum. After the adsorption is completed, the in-situ drift spectrum is then measured as the temperature rises. NH3 has introduced again, and the corresponding spectra were recorded as a function of time. The reactivity of the pre-adsorbed NO + O2 with the NH3 species was investigated for the sample 5 at 200 ℃using in-situ DRIFTS, and the spectra were recorded according to time. The results are shown It can be seen from Fig. 3 that the adsorption peaks of NOX species in the range of 1000–2000 cm-1 are analyzed, of which 1290 cm-1 and 1132 cm-1 belong to the adsorption peak of the L-acid coordination NH3 species and at 1445 cm-1 . The peak is attributed to NH4+ at the Brønsted acid site. There was no significant change in the intensity of the absorption peak of the bidentate nitrate formed at 1549 cm-1 throughout the reaction, indicating that bidentate nitrate did not react with NH3. The peak of NH2 species formed by adsorption of NH3 on the catalyst surface at 1549 cm-1 was detected, so it is understood that during the NO +O2pre-adsorption process, the remaining adsorbed oxygen on the surface promotes the dehydrogenation of NH3 to form the active intermediate NH2. In summary, it is concluded that monodentate nitrates can react with L-acid sites NH3 and NH2.
Moreover, authors stated that the catalysts were outgassed prior to the TPD and TPR runs. What was the conditions of the outgassing?
Response:
And the samples were degassed with Ar gas at 20 L / min for 30 minutes before running NH3-TPD and H2-TPR.

Reviewer 2 Report
The authors have included some explanations and discussions of the data.
Author Response
The authors have included some explanations and discussions of the data.
Response:
I have received your comments on the paper, and thank you very much for your recognition of our work and articles. We look forward to working with you next time.
Reviewer 3 Report
Thanks for the clarifications.
One final thing to do - in 2.2 please state what XRF instrument you used.
Author Response
Dear Editors and Reviewers:
Thank you for your letter and for the reviewers’ comments concerning our manuscript entitled “Surface Properties and Denitrification Performance of Impurity-Removed Rare Earth Concentrate” (ID: materials-688919).Those comments are all valuable and very helpful for revising and improving our paper, as well as the important guiding significance to our researches. We have studied comments carefully and have made correction which we hope meet with approval. The main corrections in the paper and the responds to the reviewer’s comments are as flowing:
Responds to the reviewer’s comments:
One final thing to do - in 2.2 please state what XRF instrument you used.
Response:
Thank you very much for your reminder, we have updated the information.
And X-ray fluorescence analysis(U-2200 RoHS Heavy Metal Detection Spectrometer)
We appreciate for Editors and Reviewers’ warm work earnestly, and hope that the correction will meet with approval.
Once again, thank you very much for your comments and suggestions.
Thank you and best regards.
Yours sincerely,
Wen-fei Wu
Corresponding author: Wen-fei Wu
Name: Wen-fei Wu
mail: [email protected] (Prof. Wen-fei Wu)